# Cytotoxicity and Antimicrobial Resistance of *Aeromonas* Strains Isolated from Fresh Produce and Irrigation Water

**DOI:** 10.3390/antibiotics12030511

**Published:** 2023-03-03

**Authors:** Alberto Pintor-Cora, Olga Tapia, María Elexpuru-Zabaleta, Carlos Ruiz de Alegría, Jose M. Rodríguez-Calleja, Jesús A. Santos, Jose Ramos-Vivas

**Affiliations:** 1Research Group on Foods, Nutritional Biochemistry and Health, Universidad Europea del Atlántico, 39001 Santander, Spain; 2Department of Food Hygiene and Food Technology, Veterinary Faculty, Universidad de León, 24071 León, Spain; 3Service of Microbiology, University Hospital Marqués de Valdecilla, Avda. Valdecilla s/n, 39008 Santander, Spain; 4Research Group on Foods, Nutritional Biochemistry and Health, Universidad Internacional Iberoamericana, Campeche 24560, Mexico; 5CIBER of Infectious Diseases-CIBERINFEC, Instituto de Salud Carlos III, 28029 Madrid, Spain

**Keywords:** *Aeromonas*, virulence, cytotoxicity, antimicrobial resistance, host–pathogen interactions

## Abstract

The genus *Aeromonas* has received constant attention in different areas, from aquaculture and veterinary medicine to food safety, where more and more frequent isolates are occurring with increased resistance to antibiotics. The present paper studied the interaction of *Aeromonas* strains isolated from fresh produce and water with different eukaryotic cell types with the aim of better understanding the cytotoxic capacity of these strains. To study host-cell pathogen interactions in *Aeromonas*, we used HT-29, Vero, J774A.1, and primary mouse embryonic fibroblasts. These interactions were analyzed by confocal microscopy to determine the cytotoxicity of the strains. We also used *Galleria mellonella* larvae to test their pathogenicity in this experimental model. Our results demonstrated that two strains showed high cytotoxicity in epithelial cells, fibroblasts, and macrophages. Furthermore, these strains showed high virulence using the *G. mellonella* model. All strains used in this paper generally showed low levels of resistance to the different families of the antibiotics being tested. These results indicated that some strains of *Aeromonas* present in vegetables and water pose a potential health hazard, displaying very high in vitro and in vivo virulence. This pathogenic potential, and some recent concerning findings on antimicrobial resistance in *Aeromonas*, encourage further efforts in examining the precise significance of Aeromonas strains isolated from foods for human consumption.

## 1. Introduction

The genus *Aeromonas* has received constant attention from different areas such as aquaculture and veterinary medicine [1,2,3], food safety [4,5], and clinical human microbiology [6,7]. Human infections by *Aeromonas* are more frequent and these strains have an increasing resistance to antibiotics [8,9]. Therefore, it is not only necessary to better understand antibiotic resistance in these species, but their virulence and pathogenicity mechanisms [10] as well. In addition to a growing resistance to antibiotics, strains have been detected with a tendency to resist the chlorination of water when they form biofilms, which has led the Environmental Protection Agency to place these microorganisms on a list of “Candidate Contaminants”, and to control water samples to monitor the presence of these pathogens. According to the last edition of the Bergey’s Manual of bacterial taxonomy, the genus *Aeromonas* comprises 36 species, a number that has been growing and changing during the last three decades. Correctly identifying many species is still very problematic due to the variability of some phenotypes, especially when only biochemical tests or semi-automatic systems are used for their classification [11]. In recent years, the number of scientific publications that researched the presence of *Aeromonas* strains in food has increased. Especially important is the presence of these bacteria in fresh foods, in minimally processed ready-to-eat seafoods and vegetables [4,12,13], and in water [14,15,16,17,18]. Therefore, bacteria of this genus are continually being branded as “emerging foodborne pathogens”. Another complex aspect of this bacterial genus is its virulence. *Aeromonas* strains can produce an array of virulence factors that most published papers have shown to be multifactorial, highlighting how some toxins seem to play a more relevant role in pathogenesis [19,20]. These issues are increasingly attracting researchers trying to understand the infecting bacterial biology of this genus. However, most studies on virulence are descriptive or presumptive, in which those genes supposedly involved in virulence are detected by PCR, or in which no relevant models have been used to demonstrate the pathogenicity of the strains [21,22,23]. For example, although some studies were conducted on the interaction of *Aeromonas* with host cells, we do not know much about the role that immune cells, i.e., macrophages and neutrophils, play in infections by these pathogens [24,25,26]. *A. hydrophila* and *A. salmonicida* have also been mostly used to understand the virulence factors and pathogenicity within the genus *Aeromonas*, using fish and mice as infection models. In addition, only a few other species, such as *A. dhakensis*, *A. sobria*, *A. caviae*, and *A. veronii*, have actually been studied in terms of their virulence and pathogenicity factors [25,27,28,29].

Improvements in taxonomic identification and the discovery of new *Aeromonas* species make it necessary to have a wide range of infection assays that can be used with these and other species. The present study used different cell models to study the host–pathogen interaction in different strains of *Aeromonas* isolated from food. Cell models and uncomplicated animal models helped to perform rapid screenings for the presence of functional toxins in the strains studied. They also increased our knowledge about the biology of infection of these pathogens. The present paper studied the antimicrobial susceptibility of *Aeromonas* strains isolated from fresh produce and irrigation water, and their interaction with different eukaryotic cell types, to better understand the cytotoxic capacity of these strains.

## 2. Results

### 2.1. Cytotoxic Effects of Aeromonas Strains on Several Cell Lines

*Aeromonas* spp. cytotoxicity was tested on four mammal cell lines by confocal microscopy. We first studied the interaction of *Aeromonas* strains with the human HT-29 colon cell line. Two of the strains, CI21E and AG29E1, presented high cytotoxic activity at 3 h after infection. After 90 min of incubation with these strains, the HT-29 cell monolayers were almost completely destroyed. Some examples are shown in Figure 1. The other *Aeromonas* strains did not exhibit any cytotoxicity.

No cytotoxicity appeared after 8 h of incubation with increasing volumes of bacterial ECPs, nor in HT-29 cells cultivated in transwell inserts. Some examples are shown in Figure 1D–F.

We used two other epithelial cell lines to check if the cytotoxic capacity present in these strains also affected other cell types (Vero, and primary mouse fibroblasts). Again, strains CI21E and AG29E1 presented strong cytotoxic activity in these cell lines. Some examples are shown in Figure 2. Only very few damaged cells were still attached on the coverslips 3 h after infection (Figure 2B,D). The other strains did not present cytotoxicity in these two non-human cell types either. No detectable phenotypic changes were observed in control cells infected with *E. coli* DH5-α. Again, strains CI21E and AG29E1 showed a rapid cytotoxic effect on immune cells, with the J774.1 macrophage monolayers being completely destroyed 3 h after infection. Macrophage viability decreased with time and depended on the MOI used. Strains CI21E and AG29E1 induced cytotoxicity as early as 60 min post-challenge, at a bacteria:cell ratio from ~50:1 to ~200:1. After 90 min of incubation with the MOIs, the monolayer was almost completely destroyed. At an MOI of 50:1, cytotoxicity occurred at a lesser speed (Figure 3). Cytotoxicity was not detected in macrophages infected for 3 h with other strains, or with *E. coli* DH5-α at MOI 200:1 used as a control. Some examples of these infections are shown in Figure 3.

### 2.2. Virulence in Galleria mellonella

*Galleria mellonella* larvae were challenged with fresh inocula of two non-cytotoxic *Aeromonas* strains and compared with larvae challenged with two cytotoxic strains. Survival was recorded every 12 h for up to 96 h. A total of 10^3^ CFUs (per larvae) from strains CI21E and AG29E1 killed >50% of larvae after 60 h, while strains that did not show cytotoxicity in vitro needed approximately 10^7^ CFU to achieve significant mortality (Figure 4). Mortality in control larvae was low and similar to that of larvae infected with the lowest doses of non-cytotoxic bacteria.

### 2.3. Antimicrobial Susceptibility Testing

All strains were found to be AmpC β-lactamase producers. All but one, AG29E1, presented an inducible mechanism. Strain AG29E1 was also found to be a coproducer of AmpC and ESBL enzymes. Results are showed in Table 1.

## 3. Discussion

*Aeromonas hydrophila* and *A. salmonicida* are the main representatives of the genus, which are common pathogens of fish [30,31,32,33,34]. Today, different strains of several *Aeromonas* species are emerging as causing diseases in humans, especially in individuals with compromised immune systems, causing mainly wound infections, gastroenteritis, bacteremia and septicemia [31,35,36,37]. In recent years, *Aeromonas* strains have also been considerably detected in different types of water and in fresh produce [38]. Due to the growing interest in these pathogens in other fields, such as human clinical microbiology and food safety, a better understanding of their virulence and pathogenicity is important. The virulence of these bacteria is mostly related to the presence of toxins. Five secretion systems have been identified (T1SS, T2SS, T3SS, T4SS and T6SS), mainly in *A. hydrophila*, where they have also been better characterized. However, their interactions with host cells have not been fully clarified.

It is interesting to use these cell models, since many of the infections caused by *Aeromonas* species result in gastroenteritis or skin infections, including necrotizing fasciitis. In addition, some authors have reported that *Aeromonas* is capable of invading epithelial cells, which could be an escape mechanism for these pathogens from the immune system, therefore facilitating their dissemination [26].

In a series of papers using epithelial cells from different origins, Freitas-Almeida and collaborators examined the adherence patterns of *Aeromonas* strains (including *A. hydrophila*, *A. sobria* and *A. caviae*) isolated from different origins. Some of those strains showed cytotoxicity in Vero cells, but did not show cytotoxicity in other cell types, such as HT-29, Caco-2, Hep-2 or T-84 cells [26,39,40]. Contrary to these results, strains AG29E1 and CI21E presented cytotoxicity in all the cell lines we tested, demonstrating that such cytotoxicity in some strains extends beyond highly sensitive cells to toxins such as the Vero cell line. This result is important, because *Aeromonas* strains have been shown to cause infections in very different tissues. As such, having representative models susceptible to being destroyed by these bacteria is desirable. In addition, strains exhibiting cytotoxicity in epithelial cells showed cytotoxicity in macrophages. To our knowledge, this is the first study to simultaneously compare the cytotoxicity of *Aeromonas* strains in epithelial cells of different origin and macrophages.

In another series of papers, Chopra and collaborators analyzed the role of effectors released by the secretion systems directly on Hela cells, HT-29 cells, and murine macrophages [41,42,43], as reviewed by Rosenzweig and Chopra [44]. The authors measured the amount of lactate dehydrogenase (LDH) released by infected cells, demonstrating that some toxins such as Act had a significant cytotoxic effect on HT-29 epithelial cells and RAW 364.7 macrophages, compared to a mutant strain in this effector [42]. However, there was no evidence of total destruction of the cell monolayer. This discrepancy could be due to the multiplicity of infection and the timing of the assays performed. For our part, we used a MOI 10 times higher (100:1), but half the infection time. Using LDH quantification in the case of cells infected by the strains would be impossible because 100% of the monolayer was destroyed. However, it would be interesting to learn about the necessary number of bacteria per cell that would be needed to start the cytotoxic effect. Furthermore, the toxin Act present in the supernatants of the *Aeromonas* cultures induced only a little cytotoxicity in the cells, while in our case no cellular detachment at all was appreciated. The authors state in their paper that the secretion of toxins could be directly related to the quorum sensing (QS) system of the bacterium. In previous studies, we have found no correlation between the production of quorum sensing molecules and the secretion of proteases in *A. hydrophila* [19].

Zhang and collaborators analyzed the infection of a strain of *A. sobria* in murine macrophages, evidencing a clear cytotoxicity at 90 min of infection using an MOI and an infection time similar to the one we used for this paper [28].

On the other hand, we did not analyze the patterns of a possible adhesion of non-toxic strains in detail. We did, however, verify whether the toxicity in the strains is due to a close contact between bacteria and cells, by using a model of infection with transwell inserts. From these experiments, we drew the conclusion that there must be close and rapid contact between the bacterium and the cell for bacterial cytotoxins to be activated, and to destroy the cell monolayer. Confocal microscopic photographs of *Aeromonas*-infected cells demonstrated that the cells exhibited a necrotic phenotype with a loss of plasma membrane integrity, and a non-fragmented nucleus, leaving red-stained cytoskeleton fragments spread over the surface of the coverslips. This morphology is reminiscent of the morphology of cells destroyed by the cytotoxic strains of other pathogenic species [45].

In another study, Snowden and coworkers used 81 strains of *Aeromonas* to study adherence to Hep-2 and Caco-2 cells, and toxicity in Caco-2 and Vero cells. Interestingly, the supernatant of some of the strains was found to result in a cytotoxic effect in both Vero cells and Caco-2 cells [24]. We have not observed cytotoxic activity in supernatants filtered from *Aeromonas* cultures, although some strains have high cytotoxicity. In pioneering work carried out during the mid-90s, Thornley and colleagues also conducted infection studies in Hep-2 and Caco-2 cells. In their research, the culture conditions for the bacteria clearly influenced not only adherence to eukaryotic cells, but also cytotoxicity. Observing cytotoxicity only occurred, however, when large numbers of bacteria came into contact with cells [25]. This clearly indicates that adhesion or cytotoxicity studies should be performed in short periods of infection, since *Aeromonas* proliferates easily in the culture media used for growing eukaryotic cells. Such an excess of bacteria can cause damage to cells and mask the action of toxins.

In a recent paper, Hoel and coworkers studied the phylogenetic relationship of more than one hundred strains of *Aeromonas* isolated from fresh sushi, characterizing some of its virulence factors at the genetic level, such as several toxins, and at the phenotypic level, such as motility and hemolytic activity [46]. However, these authors did not examine the production or action of toxins in vitro on cell cultures. This information could be interesting in understanding the production of these virulence factors in strains isolated from fresh products. Lastly, although many of the strains used by Freitas-Almeida’s team had the genes of toxins typical of *Aeromonas* spp. such as *aer*, *aerA*, *hly*, *ast* and *alt*, many did not show cytotoxicity in Vero cells [26,47]. This reinforced the idea that a simple PCR test is not enough to demonstrate the presence of active toxins in *Aeromonas*, and that although many of the strains of different species isolated from different sources carry those genes, only a small number of them produce toxins active against eukaryotic cells. One difference between our study and those of others is that we did not use monolayers of polarized cells. This difference could be key, although many other species of bacteria produce toxins regardless of the status of the epithelium they are encountered in. In addition, it is clear that the toxicity of the strains correlates with their virulence in vivo, so in vitro polarization might not be important. In any case, we believe that it would be interesting to perform infections in polarized cells in the future, and also to obtain the complete sequence of the genomes of cytotoxic and virulent strains to compare them with each other, and to compare them with non-cytotoxic and non-virulent strains.

As for infections in professional phagocytic cells, a recent paper by Fernández-Bravo and Figueras performed infections of different *Aeromonas* species in the monocytic human cell line THP-1 [48]. Regarding infections in phagocytes, these authors also measured cell damage in relation to LDH release in infected cultures. LDH release increased with infection time (up to 6 h) in all *Aeromonas* species tested. Although they measured the expression of some immune-related genes, they did not conduct microscopy studies, which provide information about the type and form of cell deaths.

The use of cells from the innate immune system to perform host-pathogen interaction studies are minor with the genus *Aeromonas*. However, understanding the interaction between these opportunistic pathogens and immune cells could provide novel insights in preventing and controlling the problems caused by these species.

Different types of animal models have been used to study the in vivo virulence of *Aeromonas* strains, including mice (*Mus musculus*), catfish (*Clarias gariepinus*, *Ictalurus punctatus*, *Hypophthalmichthys molitrix*), blue gourami (*Trichogaster trichopterus*), zebrafish (*Danio rerio*), slime mold (*Trichogaster tricopterus*) and the nematode *Caenorhabditis elegans* [20]. To our knowledge, no virulence studies have been conducted with *Aeromonas* species other than *A. veronii* using the *Galleria mellonella* model [49]. We consider that the infection model carried out in *G. mellonella* is an ideal complement in verifying the virulence of *Aeromonas* strains that exhibit high cytotoxicity in cell models. In view of our results, this model seems a viable, fast and cheap alternative in studying the virulence of *Aeromonas* spp.

Looking at the levels of resistance to different antimicrobials, most of the strains used in the present study do not appear to pose a potential danger. However, the more cytotoxic strain AG29E1 did present potentially more important enzymes from the point of view of antimicrobial resistance. An increase in the resistance of some *Aeromonas* species was reflected in recent studies [9,50,51,52]. This should alert us to remain vigilant regarding the epidemiology of resistance markers in these species. If antibiotic resistance and a high degree of virulence are combined, the combination can be very dangerous [53,54].

The presence of cytotoxic strains of *Aeromonas* in fresh foods, such as the ones presented in this study, can be a potential threat to a consumer’s health considering that these types of products are usually consumed raw. Further research is needed to measure the implications of the dissemination of these types of isolates through the food chains and their prevalence in fresh vegetables, and to elucidate the sources of contamination in order to design and apply effective control strategies. Lastly, the genomic sequencing of the strains CI21E and AG29E1, and some non-cytotoxic strains, will provide useful information on the presence of genes related to this cytotoxicity and virulence.

## 4. Materials and Methods

### 4.1. Bacterial Strains

*Aeromonas* strains used in this study are listed in Table 2. Strains were isolated from 145 fresh vegetables samples collected from local farms and markets and 24 samples of water used to irrigate plantations.

All isolates were primarily identified as *Aeromonas* spp. using selective culture media: McConkey agar (Oxoid, Hampshire, UK) supplemented with 16 µg/mL of cefoxitin, and Chromagar ESBL (ChromAgar, Oxoid, Hampshire, UK). For further identification, the Bruker Daltonics MALDI-TOF mass spectrometry device (Bruker Daltonics, Billerica, MA, USA) was used. The strains were routinely cultured in Luria broth (Thermo Fisher Scientific Inc., Waltham, MA, USA) with agar (LA) (Oxoid, Hampshire, UK) at 28 °C and frozen at −80 °C with 20% glycerol.

### 4.2. Cell Lines

HT-29 (ATCC^®^ HTB-38^TM^) human colon cells were cultured in McCoy’s 5a medium (Gibco) with 10% of heat-inactivated fetal bovine serum (FBS, Invitrogen); Vero green monkey kidney cells (ATCC^®^ CCL81.4^TM^) and J774A.1 mouse macrophages (ATCC^®^ TIB-67™) were cultured in Dulbecco’s modified Eagle’s medium (DMEM), supplemented with 10% and 2 mM L-Glutamine (Gibco). Primary mouse embryonic fibroblasts (MEF) were isolated from E13.5 mouse embryos and immortalized within 2–3 passages by transduction of simian virus 40 large T antigen (SV40LT). Retroviruses (RV) were produced by transfecting the Phoenix-Eco packaging cell line (ATCC^®^ CRL-3214) with the recombinant plasmid pBABE-SV40LT (Addgene 13970, Teddington, UK) using standard methods. MEF were infected with 0.45µm filtered supernatants containing RV-SV40 and 8 µg/mL polybrene (EMD Millipore #TR-1003-G, Merk Millipore, Darmstadt, Germany) and selected using 2 g/mL puromycin (Invivogen #QLL-34-03A, Toulouse, France) for 3 days. MEF were maintained and cultured in Dulbecco´s Modified Eagle Medium (DMEM, Thermo Fisher Scientific Inc., Waltham, MA, USA) containing 4500 mg/L glucose, 4 mM L-glutamine and 1 mM sodium pyruvate (Sigma D6429, Merk Millipore, Darmstadt, Germany) and further supplemented with 10% heat inactivated fetal bovine serum (Gibco 10270-106, Thermo Fisher Scientific Inc., Waltham, MA, USA), 1X MEM containing non-essential amino acids (Gibco, 11140-035, Thermo Fisher Scientific Inc., Waltham, MA, USA) and penicillin (50 I.U./mL)/streptomycin (50 g/mL) solution (Gibco 15070-063, Thermo Fisher Scientific Inc., Waltham, MA, USA). All cell lines were grown within an incubator containing a humidified, 37 °C, atmospheric O_2_/5% CO_2_ environment.

### 4.3. Fluorescence Assays

Cells were placed in 24-well tissue culture plates containing round glass coverslips. All strains were cultured overnight in 1.5 mL of Luria broth at 27 °C with moderate shaking (120 rpm) in a VariosKan Lux multimode microplate reader (Thermo Fisher Scientific Inc., Waltham, MA, USA). Bacterial infections were performed as previously described in epithelial cells, fibroblasts and macrophages [45,55]. Bacterial suspensions were washed in phosphate-buffered saline (PBS) and adjusted to approximately 5 × 10^9^ CFU ml^−1^. Cells were infected with bacteria at a multiplicity of infection (MOI, bacterium:eukaryotic cell ratio) of ∼100:1. The infected plates were centrifuged for 4 min at 200× *g* prior to incubation to promote the adherence of bacteria to cells and to synchronize infections. Infected monolayers were then incubated at 37 °C with 5% CO_2_ for 180 min. *Escherichia coli* DH5-α at MOI 200:1 was used as a non-cytotoxic control strain. After infection, cells were washed four times and fixed with cold paraformaldehyde (3.2% in PBS) for 20 min at room temperature (RT). Cells were then permeabilized with Triton X-100 (0.1% in PBS) for 5 min at room temperature and washed five times with PBS. Atto-594 phalloidin (Sigma Aldrich, Merck KGaA, Darmstadt, Germany), which binds polymerized F-actin, was used to identify actin filaments and fibers. After infections, coverslips were mounted on glass slides with Fluoroshield containing DAPI (Sigma Aldrich, Merck KGaA, Darmstadt, Germany) to stain DNA. All preparations were examined with a Nikon A1R confocal scanning laser microscope equipped with 403 nm and 561 nm lasers. Images were captured at random with a ×40 Plan-Fluor 1.3 NA objective, and processed using NIS-Elements 3.2 software (Nikon Instruments Inc., Melville, NY, USA). All immunofluorescence experiments for each strain were repeated at least three times.

### 4.4. Cytotoxicity of Bacterial Extracellular Products

To determine the cytotoxic potential of the bacterial extracellular products (ECPs) present in the bacterial culture supernatants, bacteria were grown on LB for 24 h and collected in Eppendorf tubes by centrifugation at 7000 rpm for 5 min at RT, using a bench microcentrifuge as previously described [45]. Briefly, supernatants were sterilized by membrane filtration (0.22 µm, Merk Millipore, Darmstadt, Germany) and used immediately to challenge eukaryotic cell cultures. ECPs were added directly to the cell culture medium at different volumes (e.g., 10–500 mL, each in triplicate). Cells were incubated for periods up to 8 h and processed for confocal microscopy. Control cultures were incubated with the same volumes using fresh bacterial culture medium or cell culture medium.

### 4.5. Bacterial Cell-Contact Cytotoxicity

To test the importance of bacteria–host cell contact in cytotoxicity, we cocultured the bacteria and HT-29 cells by using transwell inserts with 0.2 µm pore size (Corning, Glendale, CA, USA) as previously described [45]. Briefly, cells were cultured in the lower chamber. Bacteria at MOI 200:1 were added in the upper chamber and incubated for 2 h. Assays were performed in duplicate in two separate experiments. Cells were also incubated with DMEM (Thermo Fisher Scientific Inc., Waltham, MA, USA) as negative control.

### 4.6. Galleria mellonella Killing Assays

*Galleria mellonella* caterpillars in the final-instar larval stage (Bichosa, Salceda de Caselas, Galicia, Spain) were stored in the dark and used within 48 h from the day of shipment. Caterpillars (250 ± 25 mg in body weight) were employed in all assays.

Two non-cytotoxic strains (LE20E and ES42E), and two cytotoxic strains (CI21E and AG29E1) were selected. Bacterial suspensions were prepared from the fresh cultures of these strains. Bacterial infection of *G. mellonella* was carried out as previously described [56]. Syringes were used to inject 10 μL aliquots of the inoculum into the hemocoel of each caterpillar via the last left proleg. Ten *G. mellonella* larvae were injected with different bacterial concentrations and placed in a 9.0 cm Petri dish lined with 8.5 cm Whatman paper, then incubated at 37 °C in the dark. Bacterial colony counts on LA were used to confirm all inocula. Larvae were individually examined for melanization, and time of death was recorded. Caterpillars were considered dead when they displayed no movement in response to touch. Assays were allowed to proceed for only 4 days, as pupa formation could occasionally be seen by day 4. Three independent replicates of each infection experiment were performed per infection strain. Two negative control groups were always prepared: one group that underwent no manipulation to control for background larval mortality (no manipulation control) and one group (uninfected control) that was injected with saline solution to control the impact of physical trauma. *G. mellonella* mortality curves were plotted using Microsoft Excel version 2210.

### 4.7. Antimicrobial Susceptibility Testing

Susceptibility testing was carried out as previously described [57] using a selection of antimicrobial agents of different categories, according to the proposal of the European Centre for Disease Prevention and Control and the Centers for Disease Control and Prevention [58], and following the indications of EUCAST (https://eucast.org/; accessed on 21 December 2022). The antimicrobial agents were Piperacillin, Cefuroxime, Cefotaxime, Ceftazidime, Cefepime, Aztreonam, Imipenem, Gentamicin, Ciprofloxacin, Sulfamethoxazole/trimethoprim, and Chloramphenicol. A MAST D72C AmpC and Extended Spectrum Beta-Lactamases (ESBL) detection kit (MAST group, Liverpool, UK) was used for ESBL confirmation, as previously described [24].

## Figures and Tables

**Figure 1 antibiotics-12-00511-f001:**
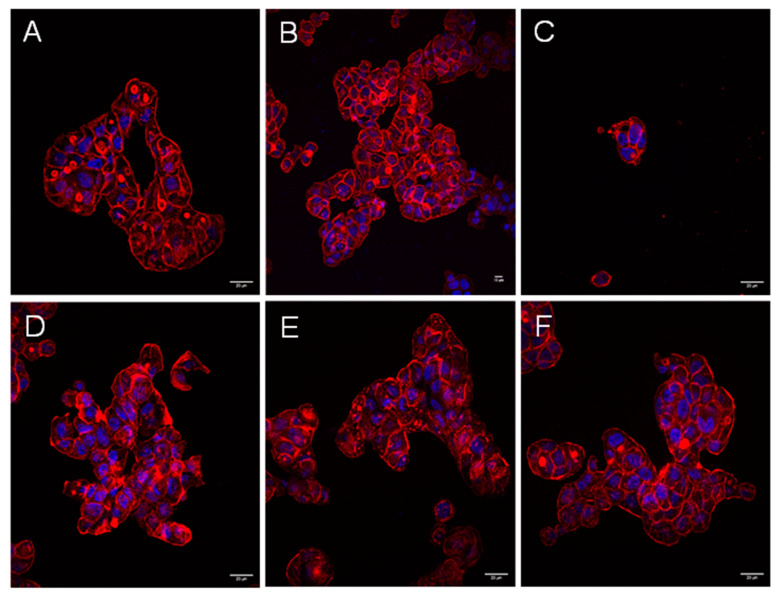
*Aeromonas* strains trigger a cytotoxic effect during the infection of HT-29 cells, but not the extracellular products they generate during their growth in the presence or absence of eukaryotic cells. Infection was carried out with non-cytotoxic strains LE20E (**A**), CI20E (**B**), and cytotoxic strain AG29E1 (**C**) for 90 and 180 min. Cells were fixed and stained for immunofluorescence. (**D**–**F**) Co-culture of *Aeromonas* strains (non-cytotoxic strain LE20E, cytotoxic strains CI21E, and AG29E1, respectively) with HT-29 cells in the inserts system. The actin cytoskeleton was labeled with Atto 594 phalloidin (red), and the nuclei stained with DAPI (blue). Results are representative of at least three independent experiments. Micrographs were originally captured at ×400 magnification. Bars indicate 20 µm.

**Figure 2 antibiotics-12-00511-f002:**
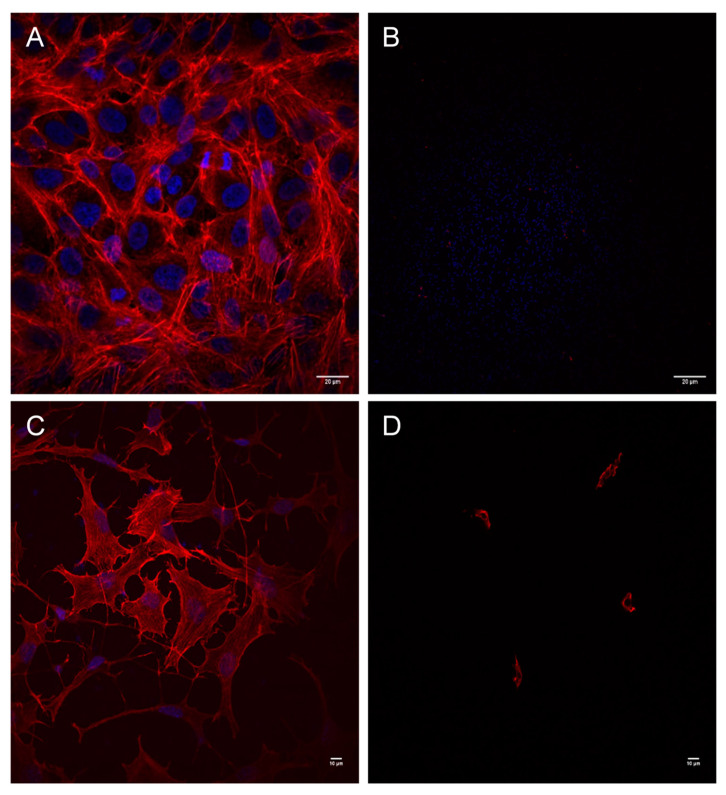
*Aeromonas* strains trigger a cytotoxic effect during infection of Vero cells and mouse fibroblasts. Infection was carried out in Vero cells with strains LE20E (**A**) and AG29E1 (**B**), and in mouse fibroblasts with non-cytotoxic strain AG26E (**C**), and cytotoxic strain AG29E1 (**D**) for 180 min. Cells were fixed and stained for immunofluorescence. The actin cytoskeleton was labeled with Atto 594 phalloidin (red), and the nuclei stained with DAPI (blue). Results are representative of at least three independent experiments. Micrographs were originally captured at ×400 magnification. Bars indicate 20 µm (**A**,**B**), or 10 µm (**C**,**D**).

**Figure 3 antibiotics-12-00511-f003:**
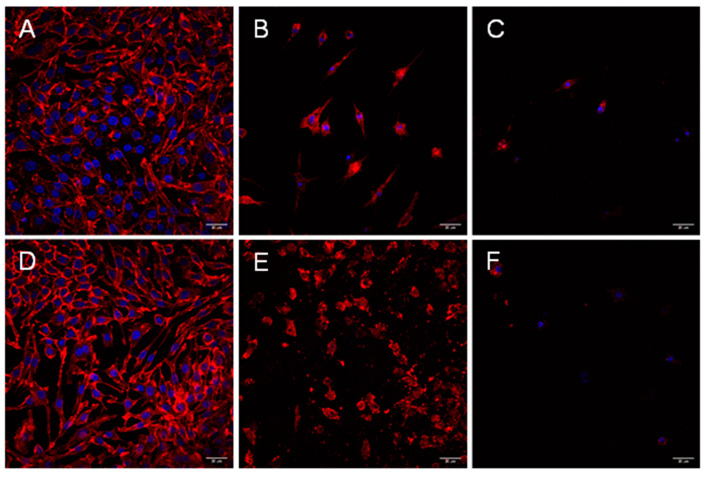
*Aeromonas* strains trigger a cytotoxic effect during infection of mouse macrophages (J774.A19). Infection was carried out in macrophages for up to 180 min with non-cytotoxic strain LE20E (**A**), cytotoxic strain CI21E (**B**), non-cytotoxic strain AG26E (**D**), and cytotoxic strain AG29E1 (**E**) for 90 min; and in CI21E (**C**) and AG29E1 (**F**) for 180 min. Cells were fixed and stained for immunofluorescence. The actin cytoskeleton was labeled with Atto 594 phalloidin (red), and the nuclei stained with DAPI (blue). Results are representative of at least three independent experiments. Micrographs were originally captured at ×400 magnification. Bars indicate 20 µm.

**Figure 4 antibiotics-12-00511-f004:**
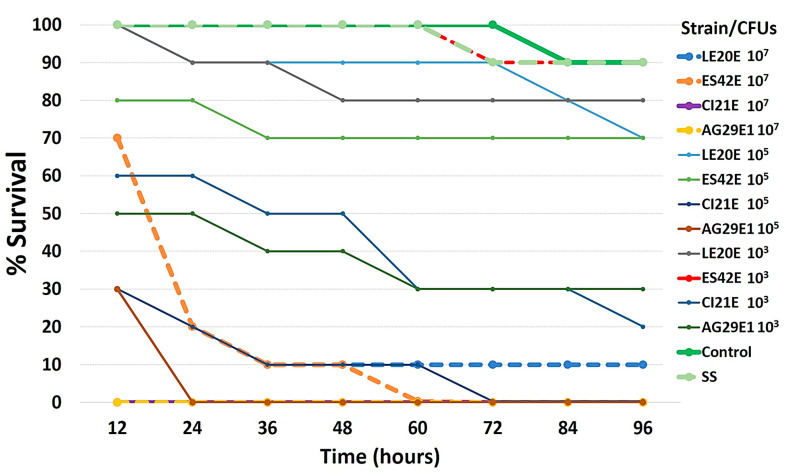
Survival rate of worms after challenge with four *Aeromonas* strains, two non-cytotoxic (LE20E and ES42E) and two cytotoxic (CI21E and AG29E1). Ten larvae were infected with saline alone, with 10^3^, 10^5^, or 10^7^ CFUs of each strain, or uninoculated (no manipulation control), incubated at 37 °C for 96 h. The time of the death of the larvae was recorded. Results are the mean of three independent experiments.

**Table 1 antibiotics-12-00511-t001:** Antimicrobial susceptibility testing in those strains used in this study.

Strain	PRL (30) *	CXM (30)	CTX (5)	CAZ (10)	FEP (30)	ATM (30)	IPM (10)	CN (10)	CIP (5)	SXT (23.75–1.25)	C (30)	AmpC/ESBL
LE20E	24.2	26.6	27.1	30	38.3	39.4	33.9	26.6	35.5	19.2	32	inducible
ES19E	17.5	19.2	28.7	30	39	40.2	35.3	28.3	40.7	22.8	32.5	inducible
ES20E	12.3	22.4	26.3	30.4	43.4	39.8	35.2	20.5	41.5	25.5	31.2	inducible
ES42E	14.8	27.4	30.5	30.2	38.4	40.8	37.2	30.5	39.5	30.2	31.5	inducible
AG26E	17.8	22.8	30.6	31.4	39.3	39.2	32.5	26.1	35.8	24.5	38.3	inducible
CI20E	22.4	33	36.1	35.3	43.3	43.3	28.4	19.3	33.3	23.2	37.1	inducible
CI21E	25.8	37.6	40.3	37.3	45.3	42.7	28.4	18.2	45.3	23.7	35.1	inducible
AG29E1	0	10.8	9.9	13.5	27.3	28.6	22.3	20.4	31.4	23.6	29.4	AmpC + ESBL

Antibiotics. PRL: Piperacillin; CXM: Cefuroxime; CTX: Cefotaxime; CAZ: Ceftazidime; FEP: Cefepime; ATM: Aztreonam; IPM: Imipenem; CN: Gentamicin; CIP: Ciprofloxacin; SXT: Sulfamethoxazole and trimethoprim; C: Chloramphenicol. * Micrograms.

**Table 2 antibiotics-12-00511-t002:** *Aeromonas* strains used in this study.

*Nº*	*Strain*	Source	*Species*
1	LE20E	Lettuce	*A. veronii*
2	ES19E	Endives	*A. hydrophila*
3	ES20E	Endives	*A. veronii*
4	ES42E	Endives	*A. veronii*
5	AG26E	Water	*A. hydrophila*
6	CI20E	Celery	*A. salmonicida*
7	CI21E	Celery	*A. salmonicida*
8	AG29E1	Water	*A. hydrophila*

## Data Availability

All strains, reagents or any other experimental data used in the article can be obtained from the corresponding author upon reasonable request.

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
