# Peer review of "Cytotoxicity and Antimicrobial Resistance of Aeromonas Strains Isolated from Fresh Produce and Irrigation Water"

_antibiotics, 2023, doi:10.3390/antibiotics12030511_

Round 1
Reviewer 1 Report
Congratulations!

Author Response
Thank you very much for the comments
Reviewer 2 Report
The authors performed their study to determine the cytotoxicity and antimicrobial resistance profiles of Aeromonas strains sourced from fresh produce and irrigation water. I have a few comments that should be addressed. Please find my comments below:
General Comments
· The number of strains is a major concern. Why did you use only eight strains in your study? Please clarify it.
· Did you isolate those eight strains or collect them from other sources or study? Please clarify it. Also, if you isolated those strains, please write briefly about the sample size, collection, and processing.
· The manuscript has a lot of grammatical errors, especially in the form of tense and punctuation. Please correct them properly.
· In the result section, please try to provide the outcomes of your study. Please don’t repeat the methodology.
· Antibiotic selection: Another major issue of your study. I wondered why you used Amoxycillin/ Clavulanic Acid, Ampicillin, Ticarcillin, and Cefoxitin. According to EUCAST guidelines (as you followed), Aeromonas species show intrinsic resistance to these antibiotics. Then why did you use them in your study? Please clarify it.
The use of these antibiotics is no sense. You should omit these antibiotics from your study.
· You should mention the name of the manufacturer, city, and country of any materials (even for the software you used) you used during your study. Please mention them properly.
· Please improve the discussion section. You have used a lot of information that shouldn’t be mentioned in the discussion section. E.g., the first paragraph of the discussion looks like another introduction to your study. You have to discuss your findings here. Discussion section means discussing your results and finding the relevance of your results by justifying your results, not mentioning all kinds of information.
Other Comments
Abstract: Results of an abstract should be in the past form. Please correct them.
Line 16: Better to use “genus” instead of “Genus.” Please correct this throughout the manuscript.
Line 18: Better to use “study” instead of “work.” Please correct this throughout the manuscript.
Line 18: “……we have studied the interaction….” should be “….we studied the interaction…..”.
Line 23: “….We have also used….” should be “…..we also used….”.
Line 38-40: Please correct the sentence.
Line 53-56: Please paraphrase this sentence. It seems complicated.
Line 57-60: Please provide a reference.
Line 60-61: Please provide a few references here.
Line 63-67: Please provide references for these statements.’
Line 70-77: The objectives of a study should be in the past form. Please correct them.
Results: Is it possible to minimize the paragraph size in the result section (especially 2.1)? Your result section seems to be an essay. If it is possible, please try to write the result section concisely.
Line 216: “G. mellonella” should be “Galleria mellonella.” A scientific name should be in full form at the start of a paragraph.
Author Response
Thank you very much for the comments. Here we write our responses to your comments and suggestions to improve the article.
Reviewer 2: The authors performed their study to determine the cytotoxicity and antimicrobial resistance profiles of Aeromonas strains sourced from fresh produce and irrigation water. I have a few comments that should be addressed. Please find my comments below:
General Comments
Reviewer 2: The number of strains is a major concern. Why did you use only eight strains in your study? Please clarify it. Did you isolate those eight strains or collect them from other sources or study? Please clarify it. Also, if you isolated those strains, please write briefly about the sample size, collection, and processing.
Authors: Aeromonas strains used in this study were isolated from 145 fresh vegetables samples collected in local farms and markets, and 24 samples of water used to irrigate plantations. We have added this paragraph in the Materials and Methods section according to the reviewer. The study was not specifically designed to search for and isolate Aeromonas strains, but once isolated, we think it is interesting to test their virulence in cell models.
Reviewer 2: The manuscript has a lot of grammatical errors, especially in the form of tense and punctuation. Please correct them properly.
Authors: The manuscript has been reviewed by a native English speaker
Reviewer 2: In the result section, please try to provide the outcomes of your study. Please don’t repeat the methodology.
Authors: Done. We have shortened the results according to reviewer's comments.
Reviewer 2: Antibiotic selection: Another major issue of your study. I wondered why you used Amoxycillin/ Clavulanic Acid, Ampicillin, Ticarcillin, and Cefoxitin. According to EUCAST guidelines (as you followed), Aeromonas species show intrinsic resistance to these antibiotics. Then why did you use them in your study? Please clarify it. The use of these antibiotics is no sense. You should omit these antibiotics from your study.
Authors: Thank you very much for these comments about the intrinsic resistance of Aeromonas. We often use these antibiotics because the EUCAST criteria are for clinical strains, not for strains of environmental origin, and so it is interesting to see that they really have intrinsic resistance. According to the reviewer, we have removed these antibiotics from Table 2.
Reviewer 2: You should mention the name of the manufacturer, city, and country of any materials (even for the software you used) you used during your study. Please mention them properly.
Authors: Done
Reviewer 2: Please improve the discussion section. You have used a lot of information that shouldn’t be mentioned in the discussion section. E.g., the first paragraph of the discussion looks like another introduction to your study. You have to discuss your findings here. Discussion section means discussing your results and finding the relevance of your results by justifying your results, not mentioning all kinds of information.
Authors: The discussion section has been shortened as much as possible, according to the reviewer.
Other Comments
Reviewer 2: Abstract: Results of an abstract should be in the past form. Please correct them.
Authors: Done
Reviewer 2: Line 16: Better to use “genus” instead of “Genus.” Please correct this throughout the manuscript.
Authors: Done
Reviewer 2: Line 18: Better to use “study” instead of “work.” Please correct this throughout the manuscript.
Authors: Done
Reviewer 2: Line 18: “……we have studied the interaction….” should be “….we studied the interaction…..”.
Authors: Done
Reviewer 2: Line 23: “….We have also used….” should be “…..we also used….”.
Authors: Done
Reviewer 2: Line 38-40: Please correct the sentence.
Authors: Done
Reviewer 2: Line 53-56: Please paraphrase this sentence. It seems complicated.
Authors: Done
Reviewer 2: Line 57-60: Please provide a reference.
Authors: Done. There are multiple references that fit well in this section. We added some references according to the reviewer
Reviewer 2: Line 60-61: Please provide a few references here.
Authors: Done. We added some references according to the reviewer
Reviewer 2: Line 63-67: Please provide references for these statements.
Authors: Done. We added some references according to the reviewer
Reviewer 2: Line 70-77: The objectives of a study should be in the past form. Please correct them.
Authors: Done.
Reviewer 2: Results: Is it possible to minimize the paragraph size in the result section (especially 2.1)? Your result section seems to be an essay. If it is possible, please try to write the result section concisely.
Authors: Done. We have shortened section 2.1 according to reviewer's comments.
Line 216: “G. mellonella” should be “Galleria mellonella.” A scientific name should be in full form at the start of a paragraph.
Authors: Done
Reviewer 3 Report
In my opinion, this research paper contains significant and relevant information that justify publication, with minor revision regarding the following issues:
1. Introduction
Page 2 lines 65-67: references are missing, please support the sentence.
2. Results
- Page 3-line 112: “Figure 1. Cytotoxicity of Aeromonas strains and ECPs on HT-29 cells.” – is this a title?. It is not a Figure 1 legend ,since that is ok near figure. Maybe is a format problem.
- Page 3-line 143: “Figure 2. Cytotoxicity of Aeromonas strains in monkey Vero cells and primary mouse fibroblasts.” – is this a title?. It is not a Figure 2 legend ,since that is ok near figure. Maybe is a format problem.
- Page 4 line 188- Figure 3. Cytotoxicity of Aeromonas strains in mouse macrophages.” – is this a title?. It is not a Figure 3 legend ,since that is ok near figure. Maybe is a format problem.
- Page 5 – line 192 -198 – Figure 3 legend – please review Figure legend, since is missing the legend (C) and (F) and (B) and ( E) are repeated.
- Page 6 – line 224 “Figure 4. Galleria mellonella killing assays..” – is this a title?. It is not a Figure 4 legend ,since that is ok near figure. Maybe is a format problem.
- Table 1 legend (page 7 – line 254) should be before Table presentation.
3. Materials and Methods,
In order to maintain coherence in the presentation, please review the reagents and culture media brands, some are missing, e.g.:
- Page 10 line 407-409:” McConkey agar supplemented with 16 μg/mL of cefoxitin, and Chromagar ESBL (ChromAgar)” please see “McConkey agar (brand reference) supplemented with 16 μg/mL of cefoxitin, (brand reference) and Chromagar ESBL (ChromAgar) (brand reference)
- Page 10 line 411: is missing the brand reference of blood agar and Luria-Bertani broth.
- Page 10 lines 432, 434 and 435, culture media brand reference are incomplete : McCoy’s 5a medium (Gibco); Dulbecco’s modified Eagle’s medium (DMEM) (brand reference missing)
- Page 11 line 464 – phalloidin (Sigma), band reference incomplete
- Page 11 line 486 – pore size (Corning) – brand reference incomplete
4. References – please review references and citation in the document: references 36 and 37 are the same (page 15 – lines 642 and 644)
Author Response
Authors: Thank you very much for your comments. Here we write our answers to your suggestions and questions to improve the article
In my opinion, this research paper contains significant and relevant information that justify publication, with minor revision regarding the following issues:
#Reviewer 3
- Introduction
Page 2 lines 65-67: references are missing, please support the sentence.
Authors: Done. We added some references according to the reviewer.
#Reviewer 3
- Results
- Page 3-line 112: “Figure 1. Cytotoxicity of Aeromonas strains and ECPs on HT-29 cells.” – is this a title?. It is not a Figure 1 legend ,since that is ok near figure. Maybe is a format problem.
- Page 3-line 143: “Figure 2. Cytotoxicity of Aeromonas strains in monkey Vero cells and primary mouse fibroblasts.” – is this a title?. It is not a Figure 2 legend ,since that is ok near figure. Maybe is a format problem.
- Page 4 line 188- Figure 3. Cytotoxicity of Aeromonas strains in mouse macrophages.” – is this a title?. It is not a Figure 3 legend ,since that is ok near figure. Maybe is a format problem.
- Page 5 – line 192 -198 – Figure 3 legend – please review Figure legend, since is missing the legend (C) and (F) and (B) and ( E) are repeated.
- Page 6 – line 224 “Figure 4. Galleria mellonella killing assays..” – is this a title?. It is not a Figure 4 legend ,since that is ok near figure. Maybe is a format problem.
Authors: Done. Thank you very much for these comments about the legends. We have fixed the problem in the text for ease of understanding.
#Reviewer 3
- Table 1 legend (page 7 – line 254) should be before Table presentation.
Authors: Done
#Reviewer 3 3. Materials and Methods,
In order to maintain coherence in the presentation, please review the reagents and culture media brands, some are missing, e.g.: - Page 10 line 407-409:” McConkey agar supplemented with 16 μg/mL of cefoxitin, and Chromagar ESBL (ChromAgar)” please see “McConkey agar (brand reference) supplemented with 16 μg/mL of cefoxitin, (brand reference) and Chromagar ESBL (ChromAgar) (brand reference) - Page 10 line 411: is missing the brand reference of blood agar and Luria-Bertani broth. - Page 10 lines 432, 434 and 435, culture media brand reference are incomplete : McCoy’s 5a medium (Gibco); Dulbecco’s modified Eagle’s medium (DMEM) (brand reference missing). - Page 11 line 464 – phalloidin (Sigma), band reference incomplete. - Page 11 line 486 – pore size (Corning) – brand reference incomplete
Authors: We have aggregated all data as far as possible, according to the reviewer.
#Reviewer 3
- References – please review references and citation in the document: references 36 and 37 are the same (page 15 – lines 642 and 644).
Authors. Thank you very much for detecting this error, the duplicate reference has been removed.
Round 2
Reviewer 2 Report
The authors addressed all of my comments. I suggest the manuscript be published in its current form. However, I have two minor comments as follows:
Line 231: still cefoxitin? You mentioned that you removed the information for those four antibiotics from the manuscript. And also penicillins?
Line 274: write "study" instead of "paper" here.
Best wishes
Author Response
Reviewer: Line 231: still cefoxitin? You mentioned that you removed the information for those four antibiotics from the manuscript. And also penicillins?
Authors: Thank you very much again for the comments and suggestions. We have corrected this sentence according to the reviewer.
Reviewer Line 274: write "study" instead of "paper" here.
Authors: Done